# Oligometastatic Gastroesophageal Adenocarcinoma: Molecular Pathophysiology and Current Therapeutic Approach

**DOI:** 10.3390/ijms21030951

**Published:** 2020-01-31

**Authors:** Jin-On Jung, Henrik Nienhüser, Nikolai Schleussner, Thomas Schmidt

**Affiliations:** Department of General, Visceral and Transplantation Surgery, University of Heidelberg, Im Neuenheimer Feld 110, 69120 Heidelberg, Germany; jin-on.jung@med.uni-heidelberg.de (J.-O.J.); Henrik.Nienhueser@med.uni-heidelberg.de (H.N.); Nikolai.Schleussner@med.uni-heidelberg.de (N.S.)

**Keywords:** gastroesophageal, adenocarcinoma, oligometastasis, metastatic mechanism, surgical resection, immunotherapy, gastric cancer, esophageal cancer

## Abstract

Gastric and esophageal cancers are dreaded malignancies, with a majority of patients presenting in either a locally advanced or metastatic state. Global incidences are rising and the overall prognosis remains poor. The concept of oligometastasis has been established for other tumor entities and is also proposed for upper gastrointestinal tract cancers. This review article explores metastasis mechanisms on the molecular level, specific to esophageal and gastric adenocarcinoma. Existing data and recent studies that deal with upper gastrointestinal tumors in the oligometastatic state are reviewed. Furthermore, current therapeutic targets in gastroesophageal cancers are presented and discussed. Finally, a perspective about future diagnostic and therapeutic strategies is given.

## 1. Introduction

Oligometastasis is a potentially treatable tumor state defined as a type of metastasis in which the primary tumor forms a small number of metastatic tumors in one or two other parts of the body [1]. The concept of oligometastasis was first proposed in 1995 by Hellman and Weichselbaum [2]. It describes a transitional state between locally limited and widespread systemic disease. The concept was revisited in 2011 by the same authors, who then distinguish true and induced oligometastasis. True oligometastasis is present when primary tumor cells have reduced metastatic competence, making the oligometastatic state potentially curable. In contrast, induced oligometastasis is a tumor state with extensive malignant capabilities, only being suppressed by systemic therapy [3].

The curative approach of treating oligometastatic tumors has already been performed successfully, for instance, in colorectal cancer (hepatic and pulmonary metastases), soft tissue cancer (pulmonary metastases), and non-small-cell lung cancer (brain metastases) [4]. However, it remains unclear to this date if a real curative approach for oligometastatic esophageal and gastric cancer exists. It is therefore necessary to deepen our understanding of the molecular pathophysiology and clinical characteristics of oligometastasis in regard to these specific tumor entities.

In addition, the topic of oligometastasis might eventually reflect an area of conflict between surgical and medical oncologists. Generally speaking, clinical features such as metachronous metastasis, patients responding to neoadjuvant therapy, a limited number of metastases, and restriction to a single organ are factors that promote consideration of surgery and resection. On the other hand, synchronous metastasis, no chemotherapy or non-responders, multiple metastases with multiple organs being affected are often referred to as palliative oncology. From the surgeon’s point of view, distinct aspects are relevant for decision-making. These are, among others, long term survival, the potential benefit of surgery as proposed in retrospective studies, resectability, control of symptoms, and quality of life. In summary, three main interests are the cornerstones of a potential conflict of interest, being (1) the oncologic meaningfulness (often represented by the oncologist), (2) the technical feasibility (represented by the surgeon), and finally (3) the patient’s expectations (see Figure 1).

To resolve these conflicts, it will be necessary to find clear definitions especially regarding the highly heterogenous localizations of possible oligometastases. Furthermore, a clear definition is needed to separate those oligometastatic states that can profit from a surgical resection from oncologic therapy, which is to eventually deliver the best possible care to the patient.

This review article elaborates on the pattern of metastatic spread for gastroesophageal cancers. The review focuses on metastasis mechanisms on the molecular level specific to upper gastrointestinal cancers. The goal is to identify the most relevant pathways and molecular patterns that may help differentiate oligometastatic tumors from aggressive tumors that do not remain in an oligometastatic state. Furthermore, existing data and recent studies will be reviewed, which deal with resections of upper gastrointestinal tumors in the oligometastatic state. Eventually, current therapeutic targets as well as future diagnostic and therapeutic strategies will be presented.

## 2. Insights into Molecular Pathology

One requirement for tumor cells to form metastases is the ability to evade their original tissue and invade a new environment. For this metastasizing process, a cascade of actions is essential: the first step of a malignant tumor cell is to displace the extracellular matrix and to break through the basement membrane. The next step is to gain access to blood vessels and to migrate to a new localization. Following migration or penetration into the new tissue [5], the last and most critical step is to initiate the cell growth in the new surrounding [6]. Only a subpopulation of the primary tumor gains access to the blood vessels and from these circulating tumor cells, an even smaller fraction (estimated <0.01%) finally succeeds in forming metastases [7].

When approaching the problem of molecular mechanisms underlying oligometastasis from esophageal and gastric adenocarcinoma, it is at first necessary to consider how the endpoint of metastasis is being measured. Most reviewed studies correlate positivity in immunohistochemistry staining with (the amount of) positive lymph nodes. However, there are also analyses correlating specific markers with poor survival, tumor size, or parameters that are not directly linked to metastasis. It is therefore advised to pay special attention to the evaluated parameters since the conclusions drawn out of the presented studies could possibly be limited. Furthermore, most cited studies refer to a potential role of distinct pathways in migration, invasion, or lymphatic metastases. However, data about specific sites of distant organ metastases beyond lymphatic affection are sparse. Last but not least, many authors do not mention explicitly which distant metastatic sites were affected in the studies.

The following section will review the data about the molecular pathogenesis of metastases deriving from esophageal and gastric cancers, referring to the most common genetic alterations which have been described so far.

### 2.1. Esophageal Adenocarcinoma

Regarding the primary tumor of esophageal cancer, there are two main types: esophageal adenocarcinoma (EAC) and esophageal squamous cell carcinoma (ESCC) [8]. As this review focuses on oligometastatic adenocarcinoma of the upper gastrointestinal tract and thus on a more advanced stage of esophageal cancer, the molecular changes of ESCC and its molecular background will not be discussed in this article, referring to the pre-existing literature [9,10].

Bargash et al. found that the **IGF2 mRNA binding protein IMP2** (or IGF2BP2/p62) is overexpressed in EAC specimens and associated with metastasis and shorter survival [11]. The authors could demonstrate that IMP2 expression increases with tumor size, metastasis, and poor prognosis. IMP2 itself is involved in cell metabolism and it has been reported that IMP2 is responsible for obesity and liver steatosis. In terms of carcinogenesis, IMP2 expression leads to elevated levels of IGF2, which can activate the signaling pathways of MAPK and Jak-STAT.

Wang et al. found that lymph node metastases were significantly increased in EAC specimens that overexpressed **Dickkopf-3 (DKK3)**, a member of the Wnt inhibitor family [12]. The authors found suggesting evidence that DKK3 functions as a tumor suppressor gene in the metastatic setting of EAC. It is hypothesized that DKK3 acts via SMAD4 and eventually influences the TGF-beta pathway. For **Dickkopf-1 (DKK1),** however, Lyros et al. have found that its oncogenic role may be independent of the Wnt/β-catenin pathway and rather acts via the Akt-mediated intracellular pathway [13]. Both studies concluded that DKK is responsible for migration, invasion, and lymph node metastasis in esophageal adenocarcinoma.

**Tumor-associated glycoprotein 72 (TAG-72)** is positively correlated with lymphatic metastasis status and with the extent of invaded lymph nodes in adenocarcinoma of the esophagogastric junction [14]. TAG-72 is a membrane mucin in the gastroesophageal mucosa that is responsible for intercellular contact in the tumor microenvironment. Xu et al. found that serum TAG-72 levels were associated significantly with lymph node ratio, suggesting it could further be utilized as a potential clinical predictor.

According to Helminen et al., the nuclear expression of **Toll-like receptor 5 (TLR5)** was significantly associated with lymph node metastases in esophageal adenocarcinoma [15]. TLR5 is capable of recognizing bacterial flagellin, and it is hypothesized that *H. pylori* recognition by TLR5 could lead to the activation of **nuclear factor-κB (NF-κB)** in gastric cancer [16]. In this context, NF-κB seems to play an important role in metastasis. Yin et al. proposed that NF-κB suppression in OE19 cancer cells (a cell line derived from gastroesophageal junction carcinoma) leads to a blockade of tumor metastasis [17]. The study investigated diallyl disulfide as a potentially novel anticancer agent that suppressed **matrix metalloproteases (MMPs)**, induced **tissue inhibitors of metalloproteinase (TIMPs),** and successively inhibited the migration and invasion of tumor cells. All in all, the data suggest that TLR5 and NF-κB are involved in the pathogenesis and dissemination of esophageal adenocarcinoma.

**Epithelial-mesenchymal transition (EMT)** describes a cellular process where adhesion features are discarded and migratory properties evolve. EMT is crucial for wound healing, embryonic development, and tumor progression [18]. Wang et al. could demonstrate that aberrant **Gli1/2** expression was significantly associated with increased EMT and AKT pathway activity in EAC cell lines [19]. Gli1 and Gli2 are transcription factors that are considered to create a positive feedback loop in **Hedgehog (Hh)**-mediated cell proliferation. Nevertheless, the findings by Wang et al. are based on laboratory results and it remains unclear if there is an association of Gli1/2 overexpression to specific metastatic sites.

It has been proposed that **cancer stem cells (CSC)** play an essential role in the mechanism of tumor metastasis [20]. To this date, CSCs have been investigated in ESCC to a much larger extent compared to EAC. For instance, Chen et al. analyzed the human esophageal cancer cell line TE-1 and found that **placental growth factor (PlGF),** as well as **MMP9**, were overexpressed in cancers with metastases compared to those without metastases. In theory, PlGF activates MMPs, which in turn breaks down the extracellular matrix and eventually facilitates metastatic spread. Furthermore, the authors demonstrated that PlGF-positive tumor cells grew significantly faster than PlGF-negative cells. Although the cancer line TE-1 is of esophageal squamous-cell origin, the study nonetheless provides important insights into the role of the VEGF family in metastasis formation. PlGF is one of six members of **VEGF family**, VEGF-A, -B, -C, -D and -E being the others [21]. It was demonstrated that high serum levels of VEGF-A and VEGF-C correlated with advanced tumor stages and lymph node metastasis in gastric cancer [22]. The angiogenic factors **HGF and follistatin** were associated with poor prognosis in esophageal cancer patients when measured in the post-(chemo)therapeutic tumor tissue [23]. Here, levels of HGF and Follistatin differed between the tumor tissue when originating either from EAC or from ESCC. In the future, there is an urgent need to investigate CSCs in EAC and their VEGF activity.

In the literature, there are various reports describing single cases of **AFP** producing esophageal cancers which lead to multiple liver metastases [24,25,26,27]. But apart from these sporadic reports and the above mentioned associations with lymphatic metastases, data is rather sparse concerning specific molecular mechanisms for metastasis from esophageal adenocarcinoma.

### 2.2. Gastric Adenocarcinoma

**Gastric adenocarcinoma (GAC)** can be subdivided according to the Laurén classification into intestinal, diffuse [28], and intermediate types. The three groups show distinct phenotypes and different prognoses [29]. Another classification system was suggested by the WHO, dividing GAC into papillary, tubular, mucinous, and poorly cohesive carcinomas [30].

Crucial steps for initiating metastasis formation are epithelial mesenchymal transition, intravasation into blood vessels, circulating tumor cell translocation, and secondary organ metastasis. Apart from these rather general concepts, distinct genetic alterations have been described for gastric adenocarcinoma that may be characteristic of invasive tumors and (mainly lymph node) metastasis. Out of the many pathways that have been already identified in gastric cancer, only those which are involved in migration and invasion of tumor cells and could thus play a decisive role in oligometastasis are discussed in this part [31].

The amplification and overexpression of **ERBB2 (or HER2/neu)** lead to several intracellular signals such as the activation of the **MAPK** signaling pathway and is very common in intestinal-type but not in diffuse-type gastric carcinomas. It was further demonstrated that HER2/neu mutations occur especially in metastatic gastric cancer [32]. Badary et al. used immunohistochemistry staining to evaluate the expression of HER2/neu in 42 gastric cancer tissue specimens [33]. The group could show that HER2/neu expression correlated with TNM status and was further associated with histological grade. Similar findings could be detected for **Ki67** and **PTEN** as well. The MAPK pathway is eventually responsible for angiogenesis, proliferation, differentiation, apoptosis, and migration of gastric cancer cells.

Ikari et al. studied gastric tumor specimens and probes from liver metastases from a total of 74 patients after oncologic resection [34]. The authors found that **TP53 mutations** were significantly higher among patients with hepatic metastases compared to patients without metastasis. The TP53 mutations were present in both the liver metastasis tissue and primary gastric tumor, suggesting an organ-specific metastasis mechanism of TP53.

The **Wnt pathway** has been reported to play an important role in the invasion and metastasis of gastric cancer. Zhang et al. studied 180 gastric cancer specimens and found that the expression of Wnt-1 (along with other genes such as beta-catenin and E-cadherin) was significantly higher compared to 30 adjacent normal tissue samples [35]. The expression levels were further related to tumor size, tumor invasive depth, and lymph node metastasis. The Wnt pathway itself is important for embryonic development and responsible for cell fate specification, proliferation, and migration [31].

Another important pathway that has been attributed to metastatic gastric cancer is the **TGF beta pathway**. In later stages of gastric cancer, TGF beta signaling increases the interaction of tumor cells with the extracellular matrix and stimulates angiogenesis and cell motility [36]. This way, TGF beta ultimately promotes invasion and metastasis of cancer cells. Two important actors in the TGF beta pathway are **RUNX3** and **SMAD4,** whose down-regulation can turn off the tumor-suppressing signals of the TGF beta pathway. In contrast, the overexpression of TGF beta was associated with lymph node metastasis and tumor progression based on in vitro studies [37]. Lin et al. were also able to detect elevated serum levels of TGF beta in gastric cancer patients with venous invasion [38].

The phase III AVAGAST trial provided evidence that angiopoietin-2 **(Ang-2)** serum levels in gastric carcinoma patients strongly correlate with hepatic metastases in particular [39]. Ang-2 is currently viewed as a driver for tumor angiogenesis independent from VEGF and it has been hypothesized that the up-regulation of Ang-2 during anti-VEGF treatment is one form of acquired tumor resistance [40]. In this context, it is essential that Ang-2 recruits myeloid cells to prepare distant metastases, especially in the lung [41], thus leading to a more aggressive metastatic form that would be incompatible with the concept of oligometastasis. Blank et al. also noted that serum Ang-2 had prognostic relevance in a single-center study. The authors further demonstrated that patients with distant metastases had lower serum levels of VEGF and follistatin than patients without metastases [42]. Moreover, high follistatin levels in the tumor tissue were significantly associated with positive lymph nodes and distant metastasis.

Zheng et al. found that **KAI1** (or CD 82), as well as **tenascin** expression, was significantly and negatively associated with liver metastases deriving from gastric adenocarcinoma [43]. While KAI1 is considered to be a tumor metastasis suppressor gene, the glycoprotein tenascin is viewed to be involved in proliferation, invasion, and angiogenesis.

**MET** is a tyrosine kinase receptor with hepatocyte growth factor **(HGF)** being its ligand. It is responsible for embryonic development, organogenesis, and wound healing. MET is amplified in approximately 2% to 10% of gastric adenocarcinomas and is correlated with tumor depth, lymph node metastasis, and unfavorable outcome [44]. Aberrant activation of the MET pathway leads to further intracellular signaling via PI3K-AKT and RAS-MAPK pathways, altogether stimulating detachment, migration, and invasion of tumor cells. However, the phase III trials RILOMET-1 and MetGastric, which investigated the monoclonal antibodies Rilotumumab (against HGF) and Onartuzumab (against MET receptor), failed to show clinical efficacy in MET-positive gastric or gastroesophageal adenocarcinoma.

## 3. Patterns of Metastatic Spread and Common Distant Metastases

The esophagus has a specific lymphatic drainage for each anatomical part with the lymphatic fluid running upwards to the neck or downwards to the celiac area leading to specific localizations of lymphatic metastases [45]. The upper part drains mainly into the deep cervical lymph nodes, whereas the middle part routes to the thoracic duct, subcarinal lymph nodes, main bronchus lymph nodes, or the paratracheal lymphatic chain [46]. The lower part normally drains into the celiac area. Hence, most of the lymph nodes affected in cancers of the lower esophagus are in this particular area. In case of high abdominal pressure or blocked lymphatic vessels, the direction of the lymphatic fluids can, however, be inverted, leading to mediastinal lymph node metastasis from tumors at the esophagogastric junction.

For **esophageal cancer**, Ai et al. described the pattern of distant organ metastases and its risk factors for a collective of 6098 patients with EAC [47]. In the study, the most frequently affected organ was determined to be the liver followed by lung, bone, and brain metastases. These results were comparable with another study examining corpses of patients with esophageal cancer [48]. Interestingly, the histological subtype predefined the localization of the metastatic spread as ESCC had a higher rate of pulmonary metastases, whereas EAC predominantly spread to the liver. It has been also reported that EAC metastasizes more frequently to the brain compared to ESCC [49].

The histological grade and the age were also risk factors with younger patients, and patients with a higher grade showed metastatic disease more often. Although the site of the distant metastases did not have a prognostic influence, the overall survival (OS) was decreased by the number of sites, reducing the expected life span by one month per metastatic site (5.0 months OS for only one site versus 2.0 months for four sites of metastases). This observation is consistent with the theoretical concept of oligometastasis, as proposed by Hellman and Weichselbaum (see above). In general, it is essential to diagnose patients with esophageal cancer as early as possible since the OS declines rapidly once the tumor has started to spread multiple metastases. The 5-year survival rate was calculated to be almost 50% in localized tumors, 25% in regional disease, and only 5% in the case of distant metastases [50].

Metastatic **gastric cancer** shows slightly different metastatic behavior. A study by Riihimäki et al. included 7559 gastric cancer patients and found that 26% of all patients had metastasis to a single site and 13% had metastasis to more than one site [51]. The most frequently affected distant sites were the liver (48%), followed by the peritoneum (32%), lung (15%) and bone (12%). The anatomical localization was a major difference in the pattern of metastatic spreading: cancers of the cardia often spread to the liver and extraperitoneal localizations like lung or bones. However, non-cardia cancers often showed peritoneal metastases. As the blood from the cardia is partially drained via the venous system around the esophagus surpassing the liver, the higher frequency of lung metastases in cardia cancer could be thereby explained [52]. The study could demonstrate that gastric cancer preferably metastasizes either on the hematogenous route or within the peritoneum. On the other hand, patients with lung metastases often also had liver metastases but no infiltration of the peritoneum. Finally, pleural and ovarian invasion were often accompanied by peritoneal metastases.

The histological subtype also differed in terms of metastatic spreading. In comparison to papillary or tubular adenocarcinoma, signet ring cell adenocarcinoma more often metastasized to the peritoneum and bones and less frequently to lung and liver, whereas mucinous adenocarcinoma more frequently spread to the pleura and mediastinum [51].

Figure 2a,b demonstrates characteristic and common distant metastases from esophageal and gastric adenocarcinoma. In addition, there are molecular pathways and actors displayed which may be responsible for organ-specific metastases.

## 4. Surgical Resection of Oligometastatic Gastroesophageal Cancers

Concurrently with the pattern of metastatic spread, surgical resection and the extent of lymphadenectomy vary depending on the localization of upper gastrointestinal tumors. The standard surgical procedure for tumors in the upper and middle esophagus is transthoracic subtotal esophagectomy with proximal resection of the stomach, gastric conduit, and high thoracic anastomosis [53,54]. In addition, lymphadenectomy of the regional thoracic and abdominal lymph nodes is recommended for all tumors. Only for cervical esophageal tumors, the cervical lymph nodes should be resected [55].

For gastroesophageal junction adenocarcinoma (AEG) type I and higher localized esophageal tumors, the standard surgical operation is subtotal esophagectomy with proximal resection of the stomach, whereas for AEG III tumors, total gastrectomy with transhiatal distal esophagectomy is performed [56]. For AEG II, there is so far no standardized procedure since both operations—subtotal esophagectomy with proximal gastrectomy and total gastrectomy with inferior esophagectomy—are currently performed. A single-center study by Blank et al. found that patients with locally advanced AEG II tumors may benefit from right **thoracoabdominal esophagectomy (TAE)** compared to **transhiatal extended gastrectomy (THG)** in terms of long-term survival [57]. However, a meta-analysis by Heger et al. analyzed eight studies with 1134 patients undergoing TAE and 1177 patients receiving THG [58]. The authors found that lymph node dissection, R0-resection rates, anastomotic leakage rates, and 30-day mortality did not differ significantly between both groups. Except for a higher rate of postoperative morbidity after TAE, the two procedures did not show any significant differences in terms of perioperative parameters. Thus, it remains unclear which surgical approach should be considered optimal for AEG II tumors.

It is recommended to choose a resection extent according to the localization and the size of the tumor. Further studies have shown that total gastrectomy is unnecessary for AEG II tumors due to low metastasis rates at the distal stomach [59,60,61]. However, Kurokawa et al. could demonstrate that for large tumors (>6.0 cm), gastrectomy should be considered. For tumors with esophageal involvement exceeding 4.0 cm, the authors recommend transthoracic subtotal esophagectomy along with lymphadenectomy of the para-esophageal, supradiaphragmatic, and posterior mediastinal lymph nodes. For esophageal involvement of less than 2.0 cm, the authors recommend a transhiatal resection with sole abdominal lymphadenectomy, while for tumors with a size between 2.0 and 4.0 cm, the resection of the para-esophageal lymph nodes should be performed either by transhiatal or transthoracic approach.

There is little data in the current literature about the topic of oligometastasis in upper gastrointestinal tumors [62]. After having searched the PubMed database and various other online databases for oligometastatic esophageal or gastric cancer, it is to be concluded that to this date, most of the published data about oligometastatic states are mainly based on case reports and retrospective analyses.

This overview of the existing literature will discuss palliative resections of gastroesophageal tumors and the most frequent oligometastatic locations. In the end, it will also cover more uncommon metastatic sites.

### 4.1. Palliative resection

For some time, the influence of primary tumor resection on overall survival in the case of upper gastrointestinal tract malignancies in **palliative situations** was controversial. Mariette et al. studied 3202 patients with advanced gastric and junctional adenocarcinoma who were all scheduled for surgery between 1997 and 2010 at 21 French medical centers [63]. Six hundred seventy-seven patients received tumoral resection although their state was defined as palliative. Their median survival of 11.9 months was significantly longer compared to 532 palliative patients (8.5 months, *p* < 0.001) who did not receive resection. Since there might have been various other factors which led to the intraoperative decision not to resect tumors, the authors performed a multivariate analysis to further examine which patients benefitted from surgery. It was possible to identify the following situations where palliative surgery was advantageous for survival: patients had to have a low ASA score (I or II) and either (1) incomplete tumor resection without metastasis or peritoneal carcinomatosis (PC), (2) only one site of solid organ metastasis without PC, or (3) localized PC without signet ring cell histology. The authors concluded that palliative surgery can be beneficial only if prognostic factors are identified and considered properly before resection.

On the other hand, the randomized and controlled **REGATTA study** by Fujitani et al. showed that gastrectomy followed by chemotherapy did not have any survival benefit compared to chemotherapy alone in advanced gastric cancer with a single non-curable factor [64]. Non-curable factors were defined as the affliction of either the liver, peritoneum, or para-aortic lymph nodes. The study’s conclusions are limited because of the heterogeneity of the so-called non-curable factors. One possible conclusion from the study might be that resection of the primary tumor without resection of the metastases should be reserved for palliative symptom control. However, the REGATTA study was not suitable for clarifying survival differences between various (oligo-)metastatic states, making it still possible that resection of distant organ metastases could be crucial for survival: with 77% in the chemotherapy group and 73% in the gastrectomy plus chemotherapy group, most patients had peritoneal metastasis, whereas only 6% and 12%, respectively, had liver metastasis.

Since oligometastasis can affect numerous organ sites with varying and highly heterogeneous extents, it is therefore necessary to further examine more specific settings besides the palliative setting. For this reason, more concrete oligometastatic sites will be discussed in the next sections.

### 4.2. Liver Metastases

Several multicenter observational studies exist in the literature which deal with resections of hepatic metastases deriving from gastric cancer. Out of these, Kinoshita et al.’s study, published in 2015, can be highlighted due to its comparatively high case number. The colleagues performed a multicenter analysis on 256 patients from five Japanese centers to study the long-term outcome after surgical resection of hepatic metastases from gastric cancer [65]. It is of note that a majority of 150 patients (58.6%) from the study had metachronous hepatic metastases, with the remaining 106 patients (41.4%) having synchronous metastases. Although the authors indicated a mean number of resected hepatic tumors of 2.0 (SD = 2.4), it is important to remark that the number of hepatic lesions for most patients was one (n = 168, 65.6%), while only 18 patients (7.0%) had three metastases and 26 patients (10.2%) four or more. The authors stated that clinically resectable metastases were rare, although strict and careful patient selection is essential and could eventually lead to long-term survival, provided that R0 surgical resection could be achieved. Their multivariate analysis identified serosal invasion of the primary gastric cancer, multiple liver metastases (three or more), and a metastasis diameter of more than 5 cm as independent predictors for poor survival. It would have been interesting to perform a subgroup analysis on the 168 patients with solitary hepatic metastases to identify significant clinicopathological factors for survival in this specific group.

The one-, three- and five-year survival rates, median survival, and independent survival parameters of metastasectomized patients are summarized in Table 1 based on the works of Kinoshita et al. and further groups.

In this selection of most recent studies, it is of note that survival data shows a wide range as well as varying prognostic parameters. This may be due to the heterogeneous settings with a positive selection bias also being mentioned in some articles. Martella et al. published the results of their meta-analysis in 2015, including 11 observational studies which investigated surgical or local treatment of liver metastases also originating from gastric cancer [71]. In summary, the meta-analysis demonstrated that patients who underwent the most aggressive local treatment for hepatic metastases were shown to have a significantly higher survival rate compared to patients who solely received palliation or systemic treatment.

Addressing the same clinical question, Markar et al. published a meta-analysis in 2016 [72]. Thirty-nine studies could be included, which were performed between 1990 and 2015, each enrolling at least 10 patients. The authors could confirm that the resection of a solitary and unilobar hepatic metastatic lesion led to an additional survival benefit compared to multiple hepatic metastases (OR = 0.31, *p* = 0.011). Furthermore, the ability to achieve an R0 resection margin was an essential factor in achieving a resection of hepatic metastases which was significantly superior in survival analysis.

Both meta-analyses conclude that curative and also palliative surgery of liver metastases from gastric cancer can improve patient survival. Most of the studies found in the current literature deal with liver metastases from gastric cancer, whereas esophageal cancer is underrepresented. However, case studies and reports stating that liver resection (combined with neoadjuvant and adjuvant chemotherapy) may be feasible and beneficial in selected patients with hepatic metastases from esophageal cancer exist [73]. Although it seems reasonable to transfer these results, large-scale studies concerning hepatic metastases from (distal) esophageal cancers are needed to substantiate this notion.

### 4.3. Pulmonary Metastases

Reliable data concerning resections of pulmonary metastases from gastroesophageal adenocarcinoma are rather sparse. However, there are plenty of case reports and small series of highly selected patients involving small case numbers. It is therefore necessary to provide an overview of the largest studies and reviews that currently exist. Aurello et al. reviewed 10 studies published between 1998 and 2013 which included a total of 44 patients, and all dealt with diverse pulmonary metastases from gastric cancer [74]. It is striking that four out of ten included studies were case reports and only two studies included ten or more patients. Likewise, the data availability and quality have to be classified as low, considering that, for example, in 62% of all cases, it was not possible to retrieve the location of the metastatic pulmonary lesion. Furthermore, 6 out of the 44 patients included in the review had multiple metastases. The authors concluded that lung metastasectomy has no standardized role in the management of pulmonary metastasized gastric cancer at the time of publication. The two largest studies from this review are listed in Table 2 underneath Aurello et al., along with other studies.

The largest single center study was published by Kanamori et al. who analyzed 33 selected patients with esophageal carcinoma and pulmonary metastases which were all resected [78]. Because of the selected patient cohort, a high complete resection rate of more than 90% could be achieved and the authors eventually confirmed the safety and effectiveness of surgical resection for this particular group. Nevertheless, the median overall survival was specified as 17.9 months, which is lower in comparison to the few other studies. Moreover, patients with negative prognostic factors were shown to have high recurrence rates, making adjuvant systemic chemotherapy in conjunction with a more differentiated follow-up strategy necessary.

Altogether, it is evident to conclude that there is no vast data available for lung metastasectomy and that the time span of existing analyses covers several decades. In particular, resection of pulmonary metastases has not been studied thoroughly in the setting of upper gastrointestinal cancer as the primary tumor. If at all, most of the reviewed case reports and small studies demonstrate that solitary pulmonary metastases might be treated by curative metastasectomy, which was allegedly beneficial for most of the patients [81,82]. Nevertheless, it is clear that to this date, there is not sufficient data to prove the significance of resecting (solitary) lung metastases in upper gastrointestinal cancer. For this purpose, it will be necessary to conduct larger (multicenter) studies with randomization between resectional and non-resectional therapy arms.

### 4.4. Peritoneal Carcinomatosis

Peritoneal carcinomatosis (PC) is a dreaded form of metastasis, especially in gastric cancer with a dismal prognosis. It has been proposed that peritoneal invasion of malignant gastric cancer cells either occurs via exfoliation and direct affection of the mesothelium, iatrogenic dissemination from preceding surgery, or via lymphatic vessels [83]. Curative concepts such as cytoreductive surgery (CRS) or hyperthermic intraperitoneal chemotherapy (HIPEC) are currently not viewed as relevant for esophageal cancer [84]. For this reason, the upcoming part will focus on PC as a potentially oligometastatic state for gastric cancer. The aim is to identify and differentiate features that may turn PC either into a rather oligometastatic or into a palliative state.

A randomized phase III trial by Yang et al. [85] investigated the efficacy and safety of CRS in combination with HIPEC for the treatment of gastric cancer with PC. In the study, 68 patients with gastric cancer and peritoneal metastasis were divided into two randomized study arms, with 34 probands receiving only CRS while the other 34 probands undergoing CRS and HIPEC. The study found that synchronous peritoneal metastasis deriving from gastric cancer could be safely treated by CRS and simultaneous HIPEC, which showed to have a significantly better survival outcome (*p* = 0.046).

A 2016 study by Chia et al. analyzed 114 patients from five French centers who received CRS and HIPEC due to peritoneal metastasis from gastric cancer [86]. The group identified 81 patients with a completeness of cytoreduction score (CCS) of 0 or 1. The fact that only nine patients (11%) showed to have a disease-free survival longer than five years reaffirms the fatal nature of PC. However, all the before-mentioned patients (who were declared as “cured”) had a PCI score of less than 7 and a CCS of 0, suggesting these features are important favorable survival factors. Their relevance could be confirmed in a multivariate analysis, with better overall survival being dependent on synchronous resection of the primary tumor and PC (*p* = 0.02), a low PCI score (*p* = 0.12), and a low CCS (*p* = 0.09). All in all, the authors concluded that CRS and HIPEC might be considered in meticulously selected patients.

In this context, the **CYTO-CHIP** study by Bonnot et al. from 2019 should be pointed out. The group analyzed data from 227 patients of prospective databases at 19 French institutions who were all curatively treated for peritoneally metastasized gastric cancer and either received CRS with or without HIPEC [87]. The study used a propensity score based on inverse probability of treatment weighting (IPTW), and the matched-pairs analysis yielded a survival advantage for patients who received both CRS and HIPEC. The median survival of patients with both modalities was 18.8 months as opposed to 12.1 months for patients with CRS alone. Likewise, a statistically significant difference could be determined for 3- and 5-year overall survival as well as for 3- and 5-year recurrence-free survival. The collected data covers a timespan of 25 years, which may lead to bias due to different therapeutic protocols, depending on period and facility. Nevertheless, the study is able to deliver important conclusions on the basis of a large cohort.

The above-mentioned studies and other works are summarized in Table 3, with survival rates and median survival also demonstrated.

The natural course of gastric cancer with PC is considered to have an overall survival of less than five months, being even shorter when PC is combined with other distant metastases [90]. All cited studies provided evidence that patients who received CRS with HIPEC had a better prognosis and were thus superior compared to CRS alone. Also, it seems obvious that a low PCI and a completed cytoreduction are beneficial survival predictors. Similar to hepatic metastases, it may be possible that selected gastric cancer patients can benefit from CRS and simultaneous HIPEC, making an oligometastatic (and thus potentially better treatable) state of peritoneal carcinomatosis conceivable. Selection criteria may be well-established scores such as CCS and PCI, the possibility of complete CRS, and last but not least, synchronous metastasis of primary tumor and PC or synchronous resection of both, respectively.

### 4.5. Other Metastases

Out of 125 patients with esophageal cancer, Onal et al. identified seven patients (6%) who had solitary **brain metastases** [91]. Five out of these seven patients had an adenocarcinoma and the remaining two patients had a squamous cell carcinoma as primary tumor. The primary tumor was treated with chemoradiotherapy and was not resected in any case. The metastases were treated with neoadjuvant chemotherapy and whole-brain radiotherapy. In their study, Onal et al. demonstrated that aggressive therapy of brain metastases with either surgery, gamma-knife, or whole-brain radiotherapy could possibly improve the survival outcomes. While most of the patients died due to the progression of their primary tumor and other distant metastases, only one patient showed a progression of his brain metastasis, thus leading the authors to the conclusion that this type of oligometastasis may be a potentially treatable distant lesion, at least for symptom control.

A single case report by Inderson et al. dealt with oligometastasis in the **left adrenal gland** synchronous to a poorly differentiated adenocarcinoma of the distal esophagus [92]. Since neoadjuvant radiochemotherapy could not be extended to the oligometastatic site, the tumor board decided to perform local radiofrequency ablation (RFA) guided by endoscopic ultrasound (EUS). As final histopathology could demonstrate, the authors stated that EUS-guided RFA was safe and feasible for local tumor reduction in this case of left adrenal gland metastasis. However, it is necessary to gather more information and experience to standardize this procedure as a potentially novel way of approaching oligometastasis to the adrenal gland, especially in the preoperational setting.

From a more general point of view, the surgical resection of adrenal metastases does not seem to be associated with shorter overall survival, as a study by Howell et al. found for 62 patients with **isolated adrenal metastases from various primaries** [93]. Overall survival was not shortened, provided that the primary tumor was not any different from non-small cell lung cancer (NSCLC) and the adrenal metastasis was diagnosed metachronously. Here, a greater time interval between first diagnosis or resection of the primary and diagnosis of adrenal metastasis was favorable for survival outcome. The application of this study to gastroesophageal cancer is however limited since only three patients had an esophageal primary carcinoma and none were gastric, whereas the majority of the patients had lung and renal cancer.

### 4.6. Conclusion

It is obvious that widely acknowledged and general recommendations based on systematic analyses are missing for oligometastatic tumors of the esophagus and stomach. The German guideline, for instance, advises strictly against surgical resection in case of any known metastatic disease [94] and only approves metastasectomy if there are no metastases detected beforehand and only if small resectable lesions (such as in lung and liver) are encountered intraoperatively. However, it is tempting to propose metastasectomy in selected cases and patients since there is promising data in the reviewed literature.

Likewise, there is high controversy ongoing concerning the optimized treatment of limited metastasis (or oligometastasis, respectively). Badgwell et al. showed that surgery in the setting of metastatic disease is an uncommon approach, with a significant risk of exploration without resection [95]. It is important to note that most of the included stage IV gastric cancer patients who underwent surgery were also treated with systemic therapy and/or chemoradiotherapy preoperatively. However, it was possible to observe long-term survival in a carefully selected small subset of patients. According to Kanda et al., one of the selection variables may be the primary tumor size [96]. Based on a smaller patient collective, the study group was able to show that gastrectomy as a secondary surgery for stage IV gastric cancer patients was safe and effective for 28 patients from 12 hospitals. The enrolled patients had an outstanding complete resection rate of 92.9% and histological tumor length was identified as the only significant prognostic factor on multivariate analysis. Further parameters for proper selection of gastric cancer patients who may profit from resection at stage IV were identified by Chang et al. [97]. The authors retrospectively examined 257 cM1 gastric cancer patients, out of which 165 patients received resection and 92 did not. In uni- and multivariate analyses, the resection itself, but furthermore the number of metastatic lesions and neoadjuvant chemotherapy, were significant prognostic factors. For this reason, a surgical approach could be evaluated by considering a possible R0-resection, good physical status, solitary metastases, and the absence of peritoneal carcinomatosis. This could lead to novelties in future national guidelines [98]. In addition, treatment with new drugs such as immune checkpoint inhibitors could increase the number of patients responding to conversion therapy. At the moment, data on these treatment regimens are limited only to case reports [99] but future studies will show if oligometastatic patients benefit from these novel therapies.

After reviewing the sparse data that exists in the current literature about oligometastatic gastroesophageal cancer, it is evident that patients with metastatic tumors of the upper gastrointestinal tract generally undergo a treatment that has been, so far, highly individual. Therefore, it is necessary to deepen our understanding of the molecular mechanisms underlying metastatic spread on the pathophysiological level. In the future, it will also be necessary to perform systematic and prospective analyses by making use of large patient collectives.

## 5. Prospective Diagnostic and Therapeutic Strategies

Current national recommendations do not generally support surgical resection for stage IV esophageal cancer. In this context, the German guidelines to esophageal and esophagogastric cancer and their advice against surgical resection for any metastatic esophageal carcinoma have already been mentioned in the introduction. Likewise, the current NCCN guidelines for esophageal and esophagogastric junction cancers also only propose either systemic therapy or palliative and best supportive care for all stage IV carcinomas but not surgery [100]. Apart from locally advanced stage IVa carcinomas, all cM1, pM1, or ypM1 tumors with any T- and N-status are in this case oligometastatic tumors and classified as stage IVb esophageal adenocarcinoma according to the AJCC.

As already shown by various studies, there might be distinct clinical characteristics in patients with oligometastatic disease, which could be utilized. In this way, it may be possible to create a more elaborate and differentiated approach for patient selection to screen for patients who would profit from surgical resection. Certainly, it will be necessary to analyze these parameters in further prospective studies. However, one way of narrowing down the patient collective could be accomplished by the **preoperative prognostic score (PPS)** introduced by Blank et al. [101]. The PPS incorporates the two preoperatively available parameters: grading and clinical response after neoadjuvant therapy. Furthermore, it is necessary to either predict the possibility of R0-resection by a multidisciplinary team or evaluate the final R-status by histopathological examination. For the final R-status, the authors could demonstrate a good correlation with prognosis identifying a subgroup of patients who benefited from resection. Schmidt et al. found that patients with a favorable PPS in response to chemotherapy may be good candidates for surgical resection in metastatic esophagogastric cancer [102]. All in all, primary resection without preoperative chemotherapy is not appropriate for oligometastatic disease and the rate of R0 resection should be maximized by preoperative expert opinion. Careful patient selection could be enabled by many other important parameters that have been detected so far. Heger et al. demonstrated that neoadjuvant therapy improves outcome in locally advanced esophageal and gastric adenocarcinoma in case of signet ring cell component [103]. Furthermore, it was shown that older age is not a criterion excluding patients from surgery [104]. Future therapeutic studies should also stratify patients before neoadjuvant treatment according to their MSI status since patients with MSI-H tumors had significantly better outcomes compared to MSS tumors [105]. These aspects confirm the emerging trend of personalized and adapted treatment which should, however, be based on clear evidence.

In the previous passages about the molecular pathophysiology of metastatic spread, it was demonstrated that there are typical metastatic patterns in gastroesophageal adenocarcinoma. Nevertheless, even more specific mechanisms of oligometastatic spread should be investigated to discriminate between oligometastasis and non-curable metastasis. Beyond optimized patient selection and fundamental research, it is also necessary to discuss current and prospective strategies concerning diagnosis and treatment of oligometastatic upper gastrointestinal cancers based on available data.

The German **AIO-FLOT3** trial aimed to evaluate the oncologic outcome in patients with limited metastatic disease who received neoadjuvant chemotherapy first and then proceeded to surgical resection. It included 252 patients with resectable or metastatic gastric and gastroesophageal junction adenocarcinoma [106]. Out of 238 eligible patients, 60 were classified as having limited metastatic (or oligometastatic) disease, and again, 36 of these 60 oligometastatic tumors underwent surgical resection of the primary tumor and its metastases. The important finding of this study was that patients within the oligometastatic study arm undergoing surgery had a significantly longer progression-free survival and overall survival compared to patients who did not receive resection. This difference in survival was observable especially in patients with retroperitoneal lymph node metastases. The results are mainly consistent with the before-mentioned retrospective analysis by Schmidt et al. [102] and other studies. However, the results of the subsequent **RENAISSANCE (AIO-FLOT5)** trial should be awaited since possible selection bias could be eliminated by randomization of patients who eventually would or would not receive surgery [107]. Nevertheless, the results of the FLOT3-trial suggest surgical resection with perioperative chemotherapy for patients with oligometastatic disease in gastric and esophagogastric adenocarcinoma.

Based on these findings, Schmidt et al. have previously proposed a therapeutic algorithm which could be applied in the future for oligometastatic gastroesophageal adenocarcinoma [108]. The initial staging is recommended to be extended by MRI for liver metastases and diagnostic laparoscopy for peritoneal carcinomatosis. The protocol then envisages “induction chemotherapy” accomplished by four cycles of the FLOT regime and additional Trastuzumab in case of HER2-positivity followed by restaging in the same fashion as the initial staging. If clinical and radiologic response or further oligometastasis as a stable state is observable, surgical resection with metastasectomy could be recommended. To be precise, the authors specified, in case of liver metastases, the number of metastases to be less than three and with a diameter not greater than 3–5 cm to be eligible for resection. Thereafter, adjuvant chemotherapy is recommended with subsequent follow-up. Sisic et al. demonstrated in a propensity score matched analysis that standardized follow-up is superior to individual follow-up [109]. In the study, patients from who received standardized follow-up by the National Center for Tumor Diseases had a significantly longer overall survival (84.9 months vs. 38.4 months, *p* = 0.040). If progression or an increasing number of metastases are present, the protocol eventually recommends chemotherapy or palliative therapy [110]. An overview of the therapeutic algorithm is given in Figure 3.

Beyond the investigation of survival differences, it will also be necessary to evaluate if surgical resection can improve quality of life and lead to lower complication rates due to the oligometastatic tumor. Hopefully, the FLOT5-trial will also yield results concerning these additional aspects of patients that have been so far declared as inoperable.

Preoperative detection of metastatic lesions could be improved by pretreatment **positron emission tomography (PET)**. Erhunmwunsee et al. investigated 71 patients who underwent neoadjuvant treatment followed by esophagectomy and who had possible distant metastatic disease on standard preoperative workup. While patients with questionable liver metastases on pretreatment PET retrospectively did not benefit from surgical resection, the authors were able to show that patients with possible metastases in other sites still had a reasonable long-term survival after resection [111]. This differentiation between liver and non-liver metastases could be significant for preoperative decision-making. A possible explanation could be that liver metastases in esophageal carcinoma appear in a more advanced state and not as early as, for example, in colorectal cancer.

Moreover, liquid biopsies could be utilized to select patients with a high risk of recurrence. For instance, **circulating tumor cells (CTC)** have been discussed as a clinical tool to detect early stages, disease recurrence, and tumor progression in gastric cancer. Arigami et al. reviewed the current data to CTC for gastric cancer, concluding that there have been contradictory studies so far [112]. However, by making use of new technologies for the isolation and enrichment of tumor cells, it may be possible to improve CTC detection using novel promising markers. In general, it is also imaginable that molecular profiling of CTC gained by liquid biopsy could lead to the development of a personalized treatment [113]. Although the cited reviews deal with gastric cancer, it might be possible to transfer knowledge about hematogenous metastasis of CTC and a potential CTC-targeted therapy to esophageal carcinoma.

Currently, there are further means of tackling (oligo-)metastases that could be relevant, especially for patients with a high perioperative risk. **Radiofrequency ablation (RFA)** is an intervention that involves the insertion of an electrode into the site of metastasis to induce thermic necrosis and has already been performed successfully for many years. **Stereotactic body radiotherapy (SBRT)** offers another method of local tumor control which also has the advantage of total extracorporeal application. Kim et al. demonstrated in 2009 on a small series of nine patients with gastric cancer and para-aortic lymph node recurrence that SBRT was potentially able to salvage tumor recurrence without causing severe complications [114]. Indeed, SBRT has been investigated thoroughly for oligometastasis in other malignant diseases. Tree et al. found that SBRT is safe and effective with local control rates of about 80% [115]. Based on a collective of 321 patients treated by SBRT, Fode et al. could identify the WHO performance status, solitary metastasis, metastasis size <3 cm, metachronous metastases, and pre-SBRT chemotherapy as significant prognostic factors promoting overall survival [116]. While there is experience with SBRT available for other tumor entities, it is however important to note that few studies have dealt so far explicitly with SBRT in the setting of oligometastatic esophageal and gastric tumors. In these cases, SBRT should only be used limitedly since there is too little data for esophagogastric tumors and metastases are often located close to organs at risk, as stated by Matsushita et al. [117]. Thus, the Eastern Canadian gastrointestinal cancer consensus conference concluded that the aim of SBRT in the oligometastatic setting is limited to achieving local control and delaying progression [118]. It is important to recall that local radiotherapy is one specific option of tumor control for a limited duration, either postponing the palliative situation or leading to salvage resection. Yet more studies are needed to clarify whether the treatment of oligometastatic disease with SBRT also translates into better overall survival.

## Figures and Tables

**Figure 1 ijms-21-00951-f001:**
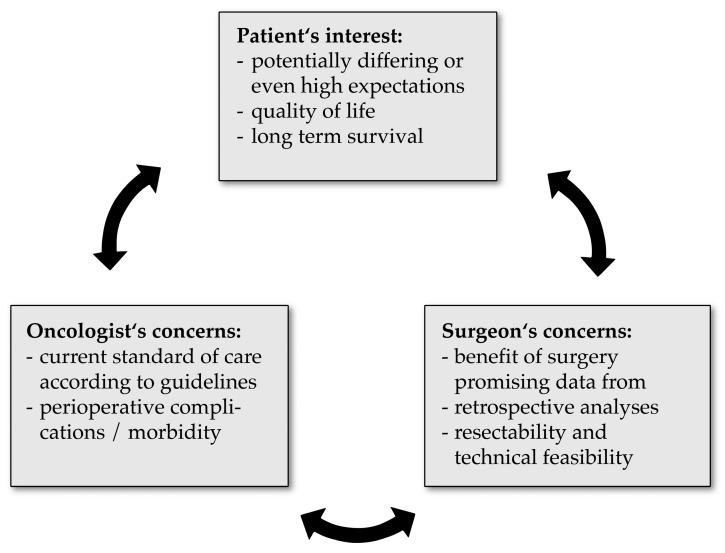
Potential area of conflict between the patient, clinical oncologist and surgeon. The major concerns are summarized underneath.

**Figure 2 ijms-21-00951-f002:**
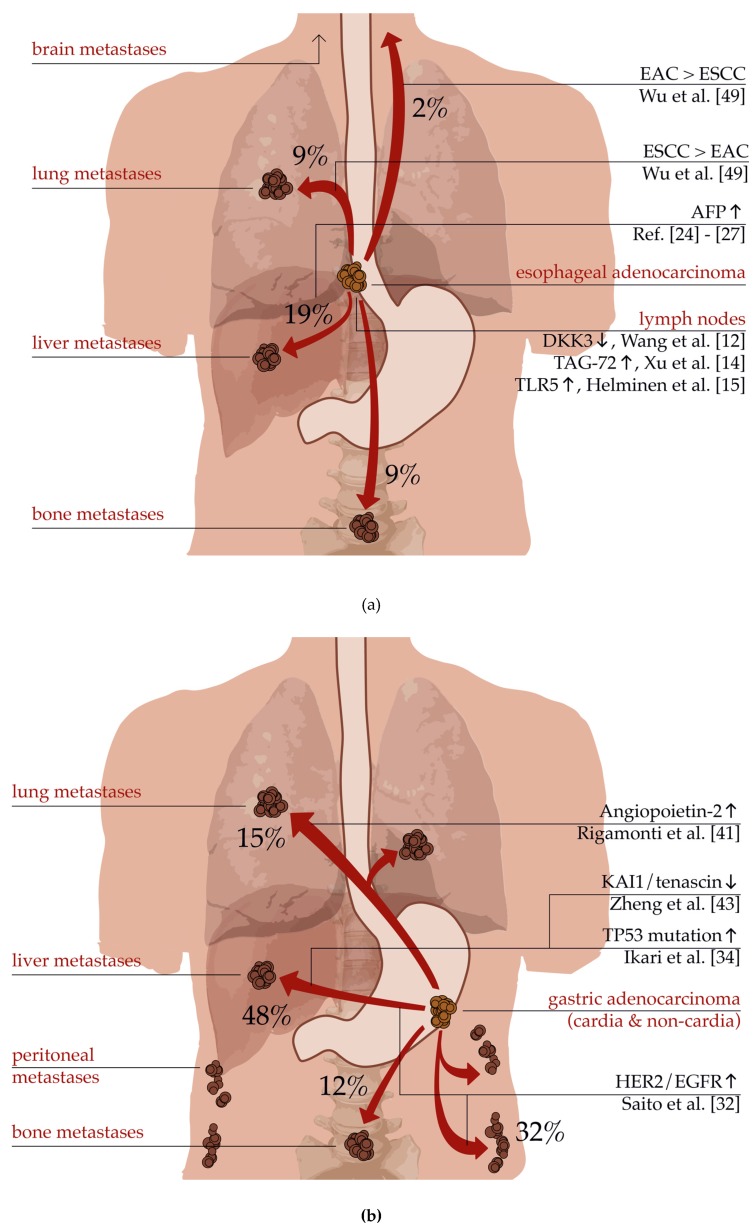
(**a**) Schematic depiction of characteristic distant metastases deriving from esophageal cancer. Frequencies according to Ai et al. [47]. (**b**) Schematic depiction of characteristic distant metastases deriving from gastric cancer. Frequencies according to Riihimäki et al. [51].

**Figure 3 ijms-21-00951-f003:**
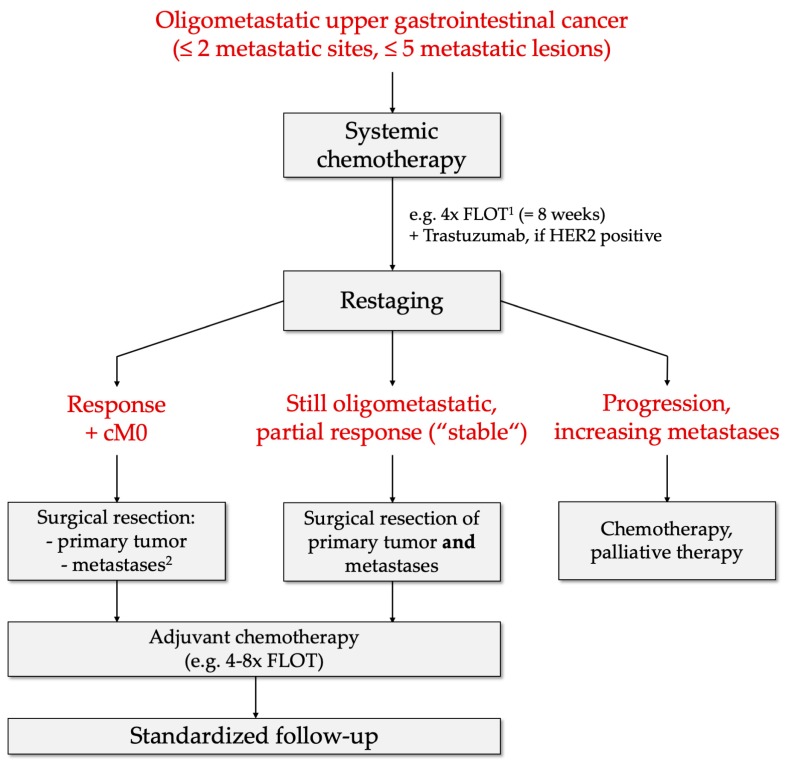
Therapeutic algorithm for patients with oligometastatic gastric or esophageal carcinoma proposed by Schmidt et al. [108]. In addition, every individual case should be discussed in interdisciplinary tumor boards. ^1^ The FLOT protocol includes: 5-Fluorouracil, Leucovorin, Oxaliplatin and Docetaxel; ^2^ based on intraoperative decision.

**Table 1 ijms-21-00951-t001:** Overview of most important ^1^ observational studies regarding resection of liver metastases from gastric cancer.

Authors	Year ^2^	n	1 Year Survival	3 Year Survival	5 Year Survival	Median Survival	Survival Factors	Ref.
Kinoshita et al.	2015	256	77.3%	41.9%	31.1%	31 mths.	serosal invasion, metastases amount, diameter of met.	[65]
Ministrini et al.	2018	144	49.9%	19.4%	11.6%	12 mths.	T4-status, H3-met., curability, recurr., no chemotherapy	[66]
Markar et al.	2016	78	64.1%	-	38.5%	-	mainly comorbidities	[67]
Oki et al.	2016	69	86.5%	51.4%	42.3%	41 mths.	solitary hep. lesions, low-grade lymph node metastases	[68]
Tiberio et al.	2016	105	58.2%	20.3%	13.1%	15 mths.	T-status, curability, met. timing, adj. chemotherapy	[69]
Takemura et al. ^3^	2013	73	71.0%	47.0%	47.0%	30.7 mths.	duration of disease-free interval	[70]

^1^ based on case numbers and recency; ^2^ year of publication; ^3^ the study investigated repeat hepatectomy exclusively.

**Table 2 ijms-21-00951-t002:** Overview of studies relevant to pulmonary oligometastases.

Authors	Year ^1^	n	1 Year Survival	3 Year Survival	5 Year Survival	Median Survival	Time Span	Ref.
Kemp et al. ^2^	2010	43	-	-	33%	29 mths.	1975–2008	[75]
Aurello et al. ^2^	2016	44	see beneath			45 mths.	1998–2013	[74]
Kobayashi et al.^2^	2013	12	-	-	58.4%	66.7 mths.	1998–2011	[76]
Yoshida et al. ^2^	2013	10	100%	100%	75% ^4^	-	2003–2012	[77]
Kanamori et al. ^3^	2018	33	79.4%	47.8%	43.0%	17.9 mths.	1992–2013	[78]
Seesing et al. ^2,3^	2019	15	67%	53%	53%	-	1991–2016	[79]
Iijima et al. ^2^	2016	10	-	30.0%	-	-	1985–2010	[80]

^1^ year of publication; ^2^ primary tumor: gastric cancer; ^3^ primary tumor: esophageal cancer; ^4^ 4-year survival.

**Table 3 ijms-21-00951-t003:** Overview of most important ^1^ studies regarding peritoneal carcinomatosis in gastric cancer.

Authors	Type	Year ^2^	Groups	n	Survival Rate	Median Survival	Prognostic Factors	Ref.
Yang et al.	Phase III RCT	2011	CRS alone	34	0% ^3^	6.5 mths.	completeness of cytoreduction, synchronous PC	[85]
			CRS+HIPEC	34	5.9% ^3^	11.0 mths.		
Bonnot et al.	multicenter, pro- and retrospective	2019	CRS alone	97	6.4% ^4^	12.1 mths.	tumor location, signet ring cell, pT, pN, low PCI, CCS	[87]
			CRS+HIPEC	180	19.9% ^4^	18.8 mths.		
Chia et al.	multicenter, retrospective	2016	CRS+HIPEC	81	18% ^4^	17.3 mths.	synchronous resection, low PCI, CCS	[86]
Boerner et al.	single center, retrospective	2016	standard ^5^	27	0% ^3^	11.0 mths.	HIPEC, age	[88]
			CRS+HIPEC	38	24.1% ^3^	17.2 mths.		
Rudloff et al.	prospective randomized trial	2014	standard ^5^	8	0% ^7^	4.3 mths.	completeness of cytoreduction, low PCI	[89]
			GYMS Arm ^6^	9	44.4%^7^	11.3 mths.		

^1^ based on recency and data availability (such as survival rates and prognostic factors); ^2^ year of publication; ^3^ three-year survival rate; ^4^ five-year survival rate; ^5^ systemic chemotherapy; ^6^ gastrectomy, CRS, HIPEC, and systemic chemotherapy; ^7^ one-year survival rate.

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
