# Peer review of "Oligometastatic Gastroesophageal Adenocarcinoma: Molecular Pathophysiology and Current Therapeutic Approach"

_ijms, 2020, doi:10.3390/ijms21030951_

Round 1

Reviewer 1 Report

I feel this review is not well organized, I would consider a more focused work.

Figure 1, 2 and 3 are not helpful, I would skip them or make them more significant. 

The concept of oligometastatic disease is about potential resectability and potential of cure. Why did you consider the part of peritoneal carcinomatosis and HIPEC? I would narrow down the review to lung and liver limited disease. The second part, about molecular pathology, is mainly a review on current treatment for metastatic disease. I would delete all the part about drugs and well known therapies. I would keep the more interesting part about patophysiology considering ONLY mechanisms linked to oligometastatic process.

This review should be synthesized better. Please keep the focus on oligometastatic disease and not on all the knowledge on gastric cancer.

Author Response

***RESPONSE TO REVIEWER 1***

Comments and Suggestions for Authors

I feel this review is not well organized, I would consider a more focused work. This review should be synthesized better. Please keep the focus on oligometastatic disease and not on all the knowledge on gastric cancer.

We thank the reviewer very much for the comments. We have re-structured the manuscript according to the suggestions and also deleted several parts to make it more focused.

Comment 1

Figure 1, 2 and 3 are not helpful, I would skip them or make them more significant.

Response

We generally appreciate the comment very much and are grateful for feedback. We have now worked on the figures and have performed minor changes to figures 1 and 3 and would propose that they remain in the manuscript since we consider them to be illustrative for the reader. Figure 2, however, underwent major changes: we now depict the metastatic routes for esophageal and gastric cancer separately (figures 2a and 2b) and with precise frequencies. Moreover, we also combine the illustration with the molecular pathways discussed in the preceding paragraph. We believe that this made the figure more significant and useful for the reader. However, we can delete these figures, if the editors still perceive them as unnecessary in the modified version.

Comment 2

The concept of oligometastatic disease is about potential resectability and potential of cure. Why did you consider the part of peritoneal carcinomatosis and HIPEC? I would narrow down the review to lung and liver limited disease. 

Response

We have included peritoneal carcinomatosis and HIPEC therapy in our review as the prominent FLOT3 trial and other publications have shown that patients with limited metastatic disease can profit from surgery. In case of localized peritoneal carcinomatosis (P1 or P2 score) the situation was treated as one form of limited disease (or oligometastatic disease) and these patients were included in study arm B. The same algorithm is being applied in the FLOT5 trial as well which focusses on the therapeutic approach of oligometastatic patients. Hence, we believe that localized peritoneal carcinomatosis is one form of oligometastasis and should be further investigated.

Comment 3

The second part, about molecular pathology, is mainly a review on current treatment for metastatic disease. I would delete all the part about drugs and well-known therapies. I would keep the more interesting part about pathophysiology considering ONLY mechanisms linked to oligometastatic process.

Response

We understand this statement and fully agree with the notion. Indeed, the molecular biology of the primary tumor has been already reviewed excessively in the preexisting literature, even on single cell sequencing level. We have thus reduced the anteceding parts about primary tumors in esophageal and gastric cancer. Furthermore, we have extended the parts about metastatic pathophysiology which are of real interest in this issue. In addition, we agree with the reviewer and have removed the paragraph concerning different drugs and well-known therapies to sustain a more focused work on oligometastasis. We realized that these aspects have been already described in the literature.

Reviewer 2 Report

This is a fairly well written article in which authors reviewed existing data about oligometastatic  upper gastrointestinal tumors, focusing on metastasis mechanisms and current therapeutic approach.

I have only some minor comments about style and typo errors:

in my opinion, this review article could benefit of a reorganization of paragraphs: paragraphs 4 and 5 could be moved after the introduction. In this way, paragraph 6 will be better joined to the more strictly clinical paragraphs 2 and 3; moreover, authors should try to trim down the articles, especially the paragraph 3; 7, line 164: it should be “patients” instead of “Patients”; 3, line 379: it should be “the” instead of “The”.

Author Response

***RESPONSE TO REVIEWER 2***

Comments and Suggestions for Authors

This is a fairly well written article in which authors reviewed existing data about oligometastatic upper gastrointestinal tumors, focusing on metastasis mechanisms and current therapeutic approach. I have only some minor comments about style and typo errors.

We thank the reviewer for this positive evaluation of our work and for the provided comments.

Comment 1

in my opinion, this review article could benefit of a reorganization of paragraphs: paragraphs 4 and 5 could be moved after the introduction. In this way, paragraph 6 will be better joined to the more strictly clinical paragraphs 2 and 3; moreover, authors should try to trim down the articles, especially the paragraph 3; 7,

Response

We thank the reviewer for this suggestion and have implemented the proposed new order of the paragraphs. We agree with you that this version is far more logical and reasonable.

Comment 2

line 164: it should be “patients” instead of “Patients”; 3, line 379: it should be “the” instead of “The”.

Response

We have performed the changes accordingly.

Round 2

Reviewer 1 Report

I accept changes and consider the review acceptable for publication.